# Prescription Patterns of Mycophenolate Mofetil in a Group of Patients from Colombia

**DOI:** 10.3390/healthcare12222224

**Published:** 2024-11-07

**Authors:** Manuel Enrique Machado-Duque, Andrés Gaviria-Mendoza, Luis Fernando Valladales-Restrepo, Álvaro Vallejos-Narváez, Natalia Piragauta-Vargas, Jorge Enrique Machado-Alba

**Affiliations:** 1Grupo de Investigación en Farmacoepidemiología y Farmacovigilancia, Universidad Tecnológica de Pereira-Audifarma S.A., Pereira 660003, Risaralda, Colombia; memachado@utp.edu.co (M.E.M.-D.); angaviria@utp.edu.co (A.G.-M.); lfvalladales@utp.edu.co (L.F.V.-R.); 2Grupo de Investigación Biomedicina, Facultad de Medicina, Fundación Universitaria Autónoma de las Américas, Pereira 660003, Risaralda, Colombia; 3Megalabs Colombia SAS, Bogotá 110911, Colombia; avallejos@megalabs.com.co (Á.V.-N.); npiragauta@megalabs.com.co (N.P.-V.)

**Keywords:** mycophenolic acid, lupus erythematosus, systemic, transplant rejections, pharmacoepidemiology

## Abstract

Background: Mycophenolate mofetil is used for the prevention of solid organ transplant rejection and for other indications, such as systemic lupus erythematosus (SLE). Objective: To determine the prescription patterns of mycophenolate mofetil in a group of Colombian patients. Methods: This was a cross-sectional study of patients receiving mycophenolate mofetil between 2021 and 2022. The data were obtained from a drug dispensing database. Sociodemographic, clinical (diagnostic), and pharmacological variables were identified. Results: A total of 979 patients who underwent treatment were identified; their mean age was 45.9 ± 17.1 years, and 87.4% were women. The main diagnosis associated with the use of mycophenolate mofetil was SLE (39.1%), followed by other rheumatic diseases (8.5%), nephrotic syndrome (7.5%), and solid organ transplantation (6.4%). The relationship between the mean dose and the defined daily dose was 0.75. Ten percent of patients received mycophenolate alone, whereas 32.9% received mycophenolate in combination therapy with conventional disease-modifying antirheumatic drugs and glucocorticoids. A total of 76.2% had polypharmacy (five or more drugs). Conclusions: Mycophenolate mofetil is used mainly in combination therapy for patients with SLE and other rheumatological diseases and for solid organ transplants at doses lower than those recommended.

## 1. Introduction

Mycophenolate mofetil, a prodrug of mycophenolic acid that is widely used for the maintenance of immunosuppressive therapies in a variety of diseases, was developed in 1995. The drug achieves immunosuppressive effects by inhibiting an enzyme involved in purine base metabolism (inosine monophosphate dehydrogenase) and preventing the synthesis of de novo guanosine nucleotides and thus the proliferation of activated T and B lymphocytes [1,2]. Mycophenolate mofetil has a bioavailability between 80.7% and 94.0%, undergoes hepatic and gastrointestinal metabolism, and is primarily eliminated through the urine [1,2].

Since its approval by the United States Food and Drug Administration and by the European Medicines Agency, mycophenolate mofetil has become the first-line drug for managing various conditions, such as prophylaxis against rejection of transplanted solid organs such as the kidney, heart, and liver [3,4], as well as the management of autoimmune hepatitis [5], lupus nephritis in patients with systemic lupus erythematosus (SLE) [6], moderate to severe psoriasis, and myasthenia gravis, among other ophthalmologic and dermatologic diseases [7,8,9].

Studies carried out in the United States have shown evidence of the effectiveness of mycophenolate mofetil over that of other drugs, such as glucocorticoids and azathioprine, in the prevention of solid organ rejection [10]. Mycophenolate mofetil has a lower incidence of adverse reactions than azathioprine does and is considered relatively safe, with minor or manageable toxicity. However, mycophenolate mofetil has been associated with various cardiovascular reactions, such as chest pain, hypertension, edema, and tachycardia. Other potential side effects include hyperglycemia, altered lipid and electrolyte profiles, anorexia, diarrhea, urinary tract infections, urticaria, anemia, leukopenia, altered liver enzymes, increased incidence of infections, and increased creatinine [11].

Studies of mycophenolate mofetil prescription patterns are scarce, and most focus on specific pathologies. Mycophenolate mofetil is used as a first-line treatment for solid organ transplant rejection prophylaxis and for other indications. According to a study by Katelaris et al. in Australia, the drug is used in SLE and other autoimmune diseases, such as vasculitis, neuroimmune disorders, polymyositis and dermatomyositis, eye and skin diseases, and various other pathologies such as autoimmune hepatitis [8]. For these diseases, the dose of mycophenolate mofetil that is usually administered is 2000 mg/day but has been described as ranging from 500 to 3000 mg/day [8].

The Colombian health system offers universal health insurance coverage to the entire population through two membership regimes (one that is contributory and paid by the employer and the worker and the other that is subsidized by the state). The system has a benefit plan that includes organ transplants and the different medications necessary to prevent their rejection (such as immunosuppressants) and medications for the treatment of different autoimmune diseases. We found no publications from Colombia showing the patterns of use of mycophenolate mofetil or its associated diagnoses; there are only general reports of its use in patients with SLE and kidney transplantation but not in patients with other indications [12,13]. Therefore, this study aimed to determine the prescription patterns of mycophenolate mofetil in a group of Colombian patients affiliated with the health system.

## 2. Materials and Methods

### 2.1. Study Design and Patients

A cross-sectional study was conducted on the prescription patterns of mycophenolate mofetil (Micoflavin^®^, Bogotá, Colombia) from a drug dispensing database. This database records information on an approximate population of 9.2 million people affiliated with four different insurers of the Colombian health system. This population represents 17.3% of the country’s population, with 85% enrolled in the contributory regime and the remaining 15% in the subsidized scheme.

Data on formulas dispensed from 1 January 2021 to 31 December 2022 were analyzed for patients who received an outpatient prescription of mycophenolate mofetil. Data on patients of either sex or city of origin who were adults (over 18 years old) at the time of receiving their first prescription during the observation period were included. No patients were excluded.

### 2.2. Variables

A database was designed to allow the following groups of variables to be collected from patients with mycophenolate prescriptions:Sociodemographic variables: sex, age, insurance scheme (contributory or subsidized), and city. The place of origin was categorized into regions of Colombia according to the classification of the National Administrative Department of Statistics (DANE) of Colombia. The regions are as follows: Caribbean Region, Central Region, Bogotá-Cundinamarca, Pacific Region, Amazon Region, and Orinoquía-Oriental Region.Clinical variables: primary diagnosis, secondary diagnosis, and comorbidities associated with the prescription. These diagnoses were obtained from the records of medical formulas according to the International Classification of Diseases, version 10 (ICD-10). For a subgroup of patients affiliated with a single insurer who has access to medical history records and prior authorization by the bioethics committee, specific data could be obtained on the type of transplant, treatment for SLE, complications, and comorbidities.Medication: mycophenolate mofetil dispensed in Colombia at all of its doses (for the quantification of the dispensation, the defined daily dose (DDD) was used as the technical unit of measurement, as recommended by the World Health Organization) was recorded. To perform this pharmaco-epidemiological study, we gathered the DDD for the main indication of the drug, the specialty of the prescriber, the interval, and the route of administration. The proportion between the mean dose and the DDD was calculated (values close to 1 imply that the defined or recommended dose is used).Concomitant medications: dispensing of drugs of interest: (a) nonsteroidal anti-inflammatory drugs (NSAIDs); (b) conventional disease-modifying antirheumatic drugs (DMARDs) (antimalarials, methotrexate, leflunomide); (c) immunosuppressive drugs (azathioprine, tacrolimus, cyclosporine, cyclophosphamide); (d) drugs of biotechnological origin; (e) glucocorticoids (prednisone, deflazacort, etc.), (f) antihypertensive, (g) antidiabetic, (h) lipid-lowering, and (i) antimicrobial agents, among others.

### 2.3. Bioethical Considerations

The protocol was endorsed by the Bioethics Committee of the Universidad Tecnológica de Pereira in the category of “research without risk” (approval code: 11-130223). The principles of confidentiality of information established by the Declaration of Helsinki were applied. In accordance with Colombian regulations, risk-free studies conducted with information from electronic records do not require obtaining informed consent or the approval of the insurer.

### 2.4. Statistical Analysis

The data were analyzed using the statistical package SPSS Statistics, version 26.0 for Windows (IBM, Armonk, NY, USA). Qualitative variables are reported herein as n (%), and quantitative variables are given as the mean (standard deviation) or median (interquartile range).

## 3. Results

### 3.1. Sociodemographic Data

A total of 979 adult patients who were prescribed mycophenolate mofetil (Micoflavin^®^) were identified during the observation period; most of these patients were females (n = 856; 87.4%), had an average age of 45.9 ± 17.1 years (range: 18–92 years), were treated mainly in the Bogotá-Cundinamarca and Caribbean regions, and were affiliated with the contributory regime of the Colombian health system.

### 3.2. Mycophenolate Mofetil

For the use of mycophenolate mofetil, the main diagnosis was SLE, followed by other rheumatic diseases, nephrotic syndrome, and solid organ transplants. The registered prescriptions were transcripts (reformulations) of the general practitioner in more than 80% of the patients, and the direct records of rheumatologists’ prescriptions were of lower doses. Table 1 shows the sociodemographic variables, the specialty of the prescribing physician, and the main diagnoses for which the drug was indicated.

The relationship between the DDD and the mean dose of mycophenolate mofetil used (1.5 g/day) was lower than that recommended by the World Health Organization (2 g/day, Anatomical Therapeutic Chemical-ATC code: L04AA06) [14]. Table 2 shows the variables related to the patterns of use, such as the average dose and dosage ranges, according to the two presentations of the drug used (250 and 500 mg) for patients overall and according to whether the primary diagnosis was SLE or a solid organ transplant. A total of 92.2% (n = 903) of the patients used the drug twice daily, and only 7.8% used it once daily. These usage patterns for the main indications identified in the study population are shown in Appendix A.

The therapeutic combination of mycophenolate mofetil together with conventional DMARDs and a glucocorticoid (n = 322, 32.9%) was found to be the most common, followed by a conventional DMARD plus an immunosuppressant and a glucocorticoid (n = 164, 16.8%) and then the combination of mycophenolate mofetil with a glucocorticoid only (n = 151, 15.4%), whereas mycophenolate mofetil was used as monotherapy only in 105 patients (10.7%). Other less frequent combinations occurred in 13.7% of the patients.

With respect to the subgroup of patients receiving 250 mg tablets (n = 56), the main specific diagnosis was solid organ transplant (n = 17; 30.4%), followed by SLE (n = 4; 7.1%), rheumatoid arthritis (n = 2; 3.6%), cardiovascular pathologies (n = 2; 3.6%), and nephrotic syndrome (n = 1; 1.8%). However, most patients taking the 250 mg tablets did not have a specific diagnosis (n = 30; 53.6%).

### 3.3. Medications

In addition to mycophenolate mofetil, an average of seven prescribed drugs were found, reaching a polypharmacy ratio of five or more drugs in 76.2% of the patients (n = 746). The medications most frequently used for the other comorbidities in this group of patients were antihypertensives, systemic glucocorticoids, proton pump inhibitors, lipid-lowering drugs, and some conventional DMARDs. Table 3 shows the frequency of use of the drugs used in patients prescribed mycophenolate mofetil.

### 3.4. Analysis of the Subgroup of Patients with Clinical Records

Information on the medical history of 22 patients receiving mycophenolate mofetil was available, and age, sex, diagnosis associated with the use of mycophenolate mofetil, and complications related to the diagnosis of solid organ transplantation (n = 17; 68.1%), SLE (n = 5; 22.7%), and the most-common comorbidities were identified (see Table 4). The mean doses, ranges, dosing intervals, records of adverse drug reactions, causes of hospitalization, and visits to the emergency services of patients are listed in Table 5.

## 4. Discussion

With this analysis of the use of mycophenolate mofetil in patients affiliated with the Colombian health system, it was possible to identify the most-common diagnoses, dosage guidelines, comorbidities, and comedications. This finding is of interest to the treating physicians of patients who mainly have SLE or have undergone solid organ transplantation, as well as to administrators, insurers, and other health service providers.

The use of mycophenolate mofetil, especially in women, is clearly related to the high frequency of SLE diagnosis (considering that women are more likely to have SLE than men are) [15]. The mean age of the patients included in this analysis coincides with that reported in a pharmacoepidemiological study in a Colombian population with SLE (approximately 42 years) [16]. This number is slightly lower than that reported for patients with lupus nephritis in Germany, whose average age was 48 years, which is probably associated with the longer duration of the disease [17]. The identification of SLE as the main diagnosis associated with the use of mycophenolate mofetil is in line with the recommendations of the clinical practice guidelines that consider it an excellent second-line option after hydroxychloroquine and glucocorticoids [18].

In a study of individuals diagnosed with SLE in Colombia, 20% of these patients were treated with mycophenolate mofetil at an average dose of 1717 mg/day, which is similar to that identified in this analysis [16]. Since the patient cohort was not monitored, this average reflects the use at a specific time and not the variations that occur, as maintenance usually starts with higher induction doses, and then maintenance is carried out with lower doses [18]. For patients diagnosed with SLE who have a significant risk of impaired kidney function or progression to lupus nephritis, ensuring adequate control of blood pressure and protection of kidney function is important. The use of angiotensin-converting enzyme inhibitors or angiotensin receptor blockers is highly recommended, as identified in this report, as is the use of sodium glucose 2 cotransporter inhibitors [18].

The use of mycophenolate mofetil for solid organ transplantation (i.e., liver transplantation) has also been reported [19], with a recommended dose of approximately 1000 mg every 12 h, usually in combination with a calcineurin or glucocorticoid inhibitor [20]. The efficacy of mycophenolate mofetil monotherapy for this purpose has also been demonstrated compared with that of calcineurin inhibitor monotherapy [21]. The mean dose of 1.5 g found for patients with organ transplantation may be justified to reduce the dose in the long term, decreasing drug tolerability problems and the risk of polyomavirus infection [22,23].

In studies conducted in the United States, more than 90% of kidney transplant patients use mycophenolate mofetil [24], following the recommendations of the Kidney Disease: Improving Global Outcomes (KDIGO) clinical practice guidelines, which recommend mycophenolate mofetil as a first-line agent at maintenance doses between 1000 mg/day in such patients who use cyclosporine concomitantly and up to 2000 mg/day for those on monotherapy [23]. The patients in this study were in the recommended dosage range, which would achieve the expected benefits of lower rejection rates and greater survival than those achieved with azathioprine alone, which is the most frequently associated immunosuppressant in the patients reported here [25,26]. In a recent study involving 23 kidney transplant patients in Columbia who used mycophenolate mofetil, only three of the patients experienced transplant rejection: two due to additional problems during the procedure and one due to intolerance to the drug at the sixth week; moreover, 13% of the patients experienced adverse reactions, mainly gastrointestinal intolerance, some of which led to a change or suspension of the drug [13].

The lack of reporting and recording of adverse reactions in this group of study patients is an additional limitation to the analysis of the use of mycophenolate mofetil. However, some visits to emergency services were identified, especially due to SARS-CoV-2 infection and diarrhea, which are recognized as common manifestations of intolerance to this drug and are frequently associated with *Giardia lamblia* or Cryptosporidium [27]. In some cases, diarrhea can lead to mycophenolate suspension, which can lead to graft rejection [27,28]. Therefore, it is necessary to implement pharmacovigilance programs that lead to the early identification of adverse reactions to avoid major complications and improve the quality of life and survival of these patients. In addition, during the COVID-19 pandemic, the use of immunosuppressants, including mycophenolate mofetil, in transplant patients increased the probability of hospitalization for causes related to this viral infection [29].

Some of the limitations identified in this study involve the insurance characteristics of the population included, which may not represent all Colombian patients. The observational and retrospective nature of the records explains the fact that some information has not been collected, and the limited access to the medical records of a small number of cases makes it difficult to identify variables related to the manner of use. All of these factors, including adherence, duration of use, preexisting clinical conditions, disease severity, and adverse reactions associated with the drug, are relevant for explaining the use of mycophenolate mofetil. Nonetheless, this investigation has several strengths, such as being designed as a pharmacoepidemiological study focused on the identification of patterns of use of mycophenolate mofetil in the Colombian population, with data from a significant number of patients identified from a robust database that has been used for multiple studies of these characteristics [30].

## 5. Conclusions

According to the findings, mycophenolate mofetil was primarily prescribed as part of combination therapy for patients with SLE, other rheumatological diseases, and solid organ (liver and kidney) transplants. The doses used were consistent with international guidelines for the main indications. There was a high frequency of concomitant prescriptions of antihypertensive agents, statins, proton pump inhibitors, and other immunomodulators and DMARDs. These results provide an initial overview of mycophenolate mofetil usage patterns in Colombia and highlight the need for further studies to gather additional information on aspects such as therapy persistence, adherence, and complications and to potentially expand the understanding of mycophenolate mofetil use.

## Figures and Tables

**Table 1 healthcare-12-02224-t001:** Sociodemographic variables, prescribing physicians, and most-common diagnoses in a group of patients using mycophenolate mofetil in Colombia, 2022.

Variable	n = 979	%
Age (mean; SD)	45.9 ± 17.0	
Female	856	87.4
Region		
Bogota-Cundinamarca	532	54.3
Caribbean	329	33.6
Central	67	6.8
Orinoquía-Oriental	26	2.7
Pacific	25	2.6
Prescribing Specialty		
General Practitioner	786	80.3
Rheumatology	84	8.6
Internal Medicine	30	3.1
Nephrology	21	2.1
Cardiology	14	1.4
Others	44	4.5
Affiliated Health System Regime		
Contributory	685	70.0
Subsidized	294	30.0
Main Diagnosis—ICD-10 ^a^		
Systemic lupus erythematosus	383	39.1
Other rheumatic diseases	83	8.5
Nephrotic syndrome	73	7.5
Solid organ transplant	63	6.4
Cardiovascular pathologies ^b^	41	4.2
Rheumatoid arthritis	21	2.1
Leukemia (all subtypes)	5	0.5
Pulmonary pathologies ^c^	5	0.5
Various other unrelated diagnoses ^d^	305	31.2

SD: standard deviation; ICD-10: International Classification of Diseases, version 10. a: The main diagnosis was the use of ICD-10 codes related to the medical prescription of mycophenolate mofetil. b: Cardiovascular pathologies: arterial hypertension, coronary heart disease, and dyslipidemia. c: Pulmonary pathologies: chronic obstructive pulmonary disease, asthma, cystic fibrosis, and interstitial disease. d: All other diagnoses that are not directly related to the use of the molecule (such as pain, skin lesions, muscular disorders) or codes directly related to the type of medical attention (i.e., general medical examination).

**Table 2 healthcare-12-02224-t002:** Patterns of the use of mycophenolate mofetil in a group of patients affiliated with the Colombian health system, 2022.

Medication	n = 979	%	Mean Dose (mg/Day)	DDD ^a^	Range (Minimum and Maximum Dose) mg/Day	Age (Mean; SD)	Female Proportion (%)
Mycophenolate mofetil	979		1563 ± 691	0.78	250–4500	45.9 ± 17.1	87.4
500 mg tablet	923	94.3	1598 ± 685	0.80	500–4500	45.5 ± 16.8	88.2
250 mg tablet	56	5.7	991 ± 513	0.49	250–2000	52.1 ± 18.8	75.0
Main indication							
Systemic lupus erythematosus	383	39.1	1753 ± 726	0.87	500–4500	38.4 ± 13.7	91.4
Solid organ transplant	63	6.4	1416 ± 594	0.71	500–3000	53.1 ± 16.6	76.2
Main combinations (mycophenolate mofetil plus the following drugs)							
Conventional DMARD + glucocorticoid	322	32.9	1712 ± 720	0.86	250–4500	38.8 ± 13.6	91.0
Conventional DMARD + immunosuppressant + glucocorticoid	164	16.8	1639 ± 715	0.82	500–4000	40.2 ± 15.5	93.3
Glucocorticoid	151	15.4	1500 ± 714	0.75	500–4000	50.7 ± 16.2	88.1
Immunosuppressant + glucocorticoid	103	10.5	1403 ± 589	0.70	500–3000	51.0 ± 18.5	81.6
Other combinations	134	13.7	1437 ± 638	0.72	500–3000	52.2 ± 17.2	79.1
Mycophenolate mofetil monotherapy	105	10.7	1400 ± 573	0.70	500–3000	56.1 ± 16.2	82.9

DMARD: disease-modifying antirheumatic drug; DDD: defined daily dose; SD: standard deviation; IQR: interquartile range. ^a^ Proportion between the mean daily dose received and the defined daily dose.

**Table 3 healthcare-12-02224-t003:** Medications used to treat comorbidities in a group of patients using mycophenolate mofetil in Colombia, 2022.

Comedications	Total Population		Systemic Lupus Erythematosus		Solid Organ Transplantation	
	n = 979	%	n = 383	%	n = 63	%
Antihypertensive agents						
ARB	489	49.9	201	52.5	29	46.0
Calcium antagonists	354	36.2	111	29.0	20	31.7
Beta-blockers	262	26.8	72	18.8	25	39.7
Mineralocorticoid receptor antagonists	258	26.4	136	35.5	4	6.3
ACEi	255	26.0	134	35.0	6	9.5
Loop diuretics	233	23.8	94	24.5	13	20.6
Thiazides	69	7.0	25	6.5	3	4.8
ARNI	17	1.7	5	1.3	1	1.6
Others for cardiovascular use						
Methyldigoxin	1	0.1	0	0.0	0	0.0
Lipid-lowering drugs						
Statins	470	48.0	180	47.0	39	61.9
Fibrates	49	5.0	12	3.1	7	11.1
Ezetimibe	9	0.9	1	0.3	0	0.0
Hematic system and others						
Acetylsalicylic acid	310	31.7	116	30.3	22	34.9
Warfarin	50	5.1	29	7.6	0	0.0
Direct oral anticoagulants	36	3.7	9	2.3	1	1.6
Clopidogrel and other antiplatelet agents P2Y12i	29	3.0	5	1.3	3	4.8
Digestive						
Proton pump inhibitors	652	66.6	227	59.3	46	73.0
Endocrinological						
Levothyroxine	230	23.5	84	21.9	14	22.2
Antidiabetic drugs						
Metformin	89	9.1	20	5.2	16	25.4
DPP-4 inhibitors	82	8.4	12	3.1	16	25.4
SGLT-2 inhibitors	67	6.8	13	3.4	7	11.1
Long-acting insulin	63	6.4	3	0.8	13	20.6
Fast-acting insulin	37	3.8	1	0.3	12	19.0
GLP-1 receptor agonists	18	1.8	1	0.3	4	6.3
Sulfonylurea	2	0.2	0	0.0	1	1.6
Psychotropic drug						
Antiepileptics	216	22.1	86	22.5	12	19.0
Atypical antipsychotics	51	5.2	17	4.4	5	7.9
Benzodiazepines	22	2.2	6	1.6	1	1.6
Conventional antipsychotics	17	1.7	7	1.8	2	3.2
Z drugs	3	0.3	0	0.0	0	0.0
Antidepressants						
Selective serotonin reuptake inhibitors	116	11.8	50	13.1	9	14.3
Atypical	98	10.0	30	7.8	8	12.7
Tricyclics	72	7.4	36	9.4	1	1.6
Selective serotonin + norepinephrine reuptake inhibitors	22	2.2	7	1.8	2	3.2
Antiparkinsonians						
Levodopa	3	0.3	0	0.0	0	0.0
Pyridostigmine/neostigmine	2	0.2	1	0.3	0	0.0
Antihistamines						
1st generation antihistamines	148	15.1	59	15.4	5	7.9
2nd generation antihistamines	179	18.3	66	17.2	9	14.3
Conventional DMARDs						
Hydroxychloroquine	320	32.7	210	54.8	1	1.6
Chloroquine	289	29.5	172	44.9	1	1.6
Methotrexate	87	8.9	28	7.3	0	0.0
Leflunomide	15	1.5	3	0.8	0	0.0
Sulfasalazine	7	0.7	3	0.8	0	0.0
Other immunomodulators						
Azathioprine	190	19.4	82	21.4	4	6.3
Tacrolimus	128	13.1	22	5.7	42	66.7
Cyclosporine	59	6.0	12	3.1	8	12.7
Everolimus	9	0.9	0	0.0	3	4.8
Sirolimus	3	0.3	0	0.0	2	3.2
Hydroxyurea	2	0.2	1	0.3	0	0.0
Biological DMARDs						
Rituximab	16	1.6	5	1.3	0	0.0
Biological DMARDs ^a^	0	0.0	0	0.0	0	0.0
Janus Kinase Inhibitors	0	0.0	0	0.0	0	0.0
Glucocorticoids						
Prednisolone	712	72.7	332	86.7	32	50.8
Methylprednisolone	78	8.0	46	12.0	1	1.6
Deflazacort	65	6.6	34	8.9	3	4.8
Hydrocortisone	13	1.3	2	0.5	1	1.6

ARB: angiotensin receptor blocker; ACEi: angiotensin-converting enzyme inhibitor; ARNI: angiotensin receptor–neprilysin inhibitor; DPP4: dipeptidyl peptidase 4; SGLT2: sodium glucose cotransporter type 2; GLP-1: glucagon-like peptide; DMARD: disease-modifying antirheumatic drug. a: Other biological DMARDs: abatacept, adalimumab, anakinra, certolizumab, etanercept, golimumab, hydroxyurea, infliximab, tocilizumab, and ustekinumab.

**Table 4 healthcare-12-02224-t004:** Description of some variables of a subgroup of 22 patients whose clinical history information was obtained.

Variables	n = 22	%
Age (median; IQR)	62.0 (45.0–71.2)	
Female	17	77.3
Mycophenolate—diagnosis associated with its use		
Liver transplant	11	50.0
Heart transplant	3	13.6
Kidney transplant	3	13.6
Systemic lupus erythematosus	5	22.7
Transplant complications		
Encephalopathy	1	5.9
Liver transplant rejection	1	5.9
Chronic kidney disease	1	5.9
Cardiomyopathy	1	5.9
Systemic lupus erythematosus		
Lupus nephritis	3	60.0
Comorbidities		
Diabetes mellitus type 2	7	31.8
Hypothyroidism	5	22.7
Dyslipidemia	3	13.6
Arterial hypertension	3	13.6
Coronary heart disease	1	4.5
Obesity	1	4.5
Prostatic hyperplasia	1	4.5
Uterine fibroids	1	4.5
Chronic kidney disease	1	4.5
Pulmonary tuberculosis	1	4.5
Atrial fibrillation	1	4.5
Osteoporosis	1	4.5

IQR: interquartile range.

**Table 5 healthcare-12-02224-t005:** Patterns of use and other variables associated with the use of mycophenolate mofetil in a subgroup of 22 patients affiliated with the Colombian health system, 2022.

Medication	n = 22	%
Mycophenolate mofetil		
Average daily dose	928 ± 372	
Dose range, mg/day	250–1500	
Dosing interval (every 12 h)	18	81.8
Dosing interval (every 8 h)	2	9.1
Median daily dose (solid organ transplant); median, IQR.	1000 (500–1000)	
Median dose/day (SLE); median, IQR.	1000 (1000–1000)	
Use prior to the observation period	17	77.3
Mycophenolate mofetil-Discontinuation	1	4.5
Mycophenolate mofetil dose change	1	4.5
Adverse reactions recorded in the medical records	0	0.0
Hospitalizations	4	18.2
Complicated COVID-19	2	9.0
Decompensated heart failure	1	4.5
Complicated urinary tract infection	1	4.5
Emergencies		
COVID-19	2	9.0
Gastroenteritis/diarrhea	2	9.0

IQR: Interquartile range; SLE: Systemic lupus erythematosus; COVID-19: Coronavirus disease.

## Data Availability

The data presented in this study are openly available in https://www.protocols.io/ at https://www.protocols.io/private/3A2E2049098A11EFA0800A58A9FEAC02, accessed on 3 May 2024.

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
