# Peer review of "Prescription Patterns of Mycophenolate Mofetil in a Group of Patients from Colombia"

_healthcare, 2024, doi:10.3390/healthcare12222224_

Round 1

Reviewer 1 Report

Comments and Suggestions for Authors

In the introduction:

- You described mycophenolate with more than needed details for your study and objective; paragraph # 3 can be shortened and merged with the previous paragraph.

- You did not describe the pattern of prescribing mycophenolate from a regional or international perspective, which limited your introduction in terms of leading to your gaps in the literature.

- I did not understand the significance of the study. You need to revise your introduction to make this clear.

In the methods:

- You did not need to mention the brand name as it is available as generic all over the world and your study did not limit your analysis to the brand version, right?

- The variables section is clear but I am not sure if it fits the journal style and at least the first section is missing (a.) for the listing of the first category of variables.

- The statistical analysis section is very limited and more inferential analyses could have been conducted.

In the results:

- For me as a non-expert in using mycophenolate, I did not understand what you mean by DDD and why is it important? This should have been described more and with more details in the methods section as it appeared multiple times in the results.

- I found Table 2 very confusing and difficult to understand in its current presentation, especially since it has many columns and rows that seem irrelevant to each other in my opinion. This needs to be revised and if needed a regression could be utilized as appropriate. Also, consider separating this into multiple tables for ease of understanding.

- "Regarding the subgroup of patients using 250 mg tablet presentations, it was found that the main specific diagnosis was solid organ transplant (n=17; 30.4%), followed by SLE (n=4; 7.1%), rheumatoid arthritis (n=2; 3.6%), cardiovascular pathologies (n=2; 3.6%) and nephrotic syndrome (n=1; 1.8%). However, most patients did not have a specific diagnosis 179 (n=30; 53.6%)." I did not understand this part as well and why it is there or why it is important.

- Table 3 is very long and does not seem to be clear enough to be informative for the readers, you need to make it clear and explain why this is important to be added.

- Tables 4 and 5 and section 3.4 are very limited and not informative and I am not sure if they can be informative noting that they only represent 2% of your sample.

In the discussion:

- What do you mean by this sentence "The use of mycophenolate mofetil, especially in women, is related to the frequency of SLE (16) diagnosis".

- This was not reflected in the results "The use of angiotensin-converting enzyme inhibitors or angiotensin 2 receptor antagonists is highly recommended, as identified in this report, as is the use of sodium-glucose 2 cotransporter inhibitors", only in the table. Also, no explanation was provided.

- I did not see in the results "The use of mycophenolate mofetil for solid organ transplantation, in this case, frequently liver transplantation has also been reported" and I did not understand how you reached it.

- How is this relevant "The use of mycophenolate mofetil for solid organ transplantation, in this case frequently liver transplantation, has also been reported in the United Kingdom (20), as the recommended dose is approximately 1000 mg every 12 hours, usually in combination with a calcineurin or glucocorticoid inhibitor (21). The efficacy of mycophenolate mofetil for this purpose has also been demonstrated compared to treatments based on calcineurin inhibitors to avoid graft rejection and patient survival for up to 5 years of follow-up (22)."?

- The discussion section needs critical revision to improve the discussion level.

Comments on the Quality of English Language

The manuscript has many English writing errors and needs revision by an editing service.

- Disease Modifying Antirheumatic Drugs (DMARDs), the term was defined many times in the manuscript which is against the scientific writing standards.

The conclusion also needs revision to reflect the study's purpose.

Author Response

Response to reviewer1

Manuscript ID: healthcare-3161455

Title: Prescription patterns of mycophenolate mofetil in a group of patients from Colombia

Thank you for considering our paper. Here, we respond point by point to the comments

In the introduction:

- You described mycophenolate with more than needed details for your study and objective; paragraph # 3 can be shortened and merged with the previous paragraph.

R/ We have shortened the information regarding mycophenolate mofetil in the first paragraph. Paragraph #3 has  also been shortened and merged with the last sentence from the previous paragraph.

- You did not describe the pattern of prescribing mycophenolate from a regional or international perspective, which limited your introduction in terms of leading to your gaps in the literature.

R/ Studies of mycophenolate mofetil prescription patterns are scarce, and most of these investigations focus on specific pathologies. We have added the previous sentence to the fourth paragraph of the introduction. We also cite one of the few studies that describe mycophenolate mofetil usage in general. Finally, we have also added a sentence that describes the usual doses.

- I did not understand the significance of the study. You need to revise your introduction to make this clear.

R/ We have revised the last paragraph in the introduction to clarify the significance of this study.

In the methods:

- You did not need to mention the brand name as it is available as generic all over the world and your study did not limit your analysis to the brand version, right?

R/ We mention the brand name in the manuscript because we included only the branded version of mycophenolate mofetil in this analysis.

- The variables section is clear but I am not sure if it fits the journal style and at least the first section is missing (a.) for the listing of the first category of variables.

R/ We have added the “a)” that was missing.

- The statistical analysis section is very limited and more inferential analyses could have been conducted.

R/ This was a descriptive study that was not intended to perform inferential analyses. We believe that other exploratory and inferential analyses would have been outside the scope of the study.

In the results:

- For me as a non-expert in using mycophenolate, I did not understand what you mean by DDD and why is it important? This should have been described more and with more details in the methods section as it appeared multiple times in the results.

R/ The explanation regarding the DDD was extended in the Section C of the variables in the methods section. The use of the DDD is standard in this type of pharmacoepidemiological study.

- I found Table 2 very confusing and difficult to understand in its current presentation, especially since it has many columns and rows that seem irrelevant to each other in my opinion. This needs to be revised and if needed a regression could be utilized as appropriate. Also, consider separating this into multiple tables for ease of understanding.

R/ We consider the table as displaying the relevant information. The figure shows the number, percentage, mean dose, proportion between the mean dose and the DDD, mean age and female proportion regarding drug use as a whole and, according to its presentation (500 and 250 mg tablets), the indications and therapies (combinations with other drugs of interest). This is a classic description of a drug usage pattern. Additionally, considering that columns are necessary for each group of rows, we believe that it is not convenient to split them into multiple tables. We also do not believe it is necessary to perform a regression analysis; we do not have a prespecified outcome, nor were we trying to make inferences.

- "Regarding the subgroup of patients using 250 mg tablet presentations, it was found that the main specific diagnosis was solid organ transplant (n=17; 30.4%), followed by SLE (n=4; 7.1%), rheumatoid arthritis (n=2; 3.6%), cardiovascular pathologies (n=2; 3.6%) and nephrotic syndrome (n=1; 1.8%). However, most patients did not have a specific diagnosis 179 (n=30; 53.6%)." I did not understand this part as well and why it is there or why it is important.

R/ This section describes the diagnoses associated with the usage of 250 mg tablets. It is important to describe because the intention of the manuscript is precisely to show how the drug is being used (presentation, dosage and indications).

- Table 3 is very long and does not seem to be clear enough to be informative for the readers, you need to make it clear and explain why this is important to be added.

R/ This table is explained in Section 3.3 of the Results. It is necessary to describe the other drugs that are used concomitantly with the medication of interest. This study helps to clarify the patterns of mycophenolate usage and how mycophenolate use is related to other therapies.

- Tables 4 and 5 and section 3.4 are very limited and not informative and I am not sure if they can be informative noting that they only represent 2% of your sample.

R/ Indeed, there are few patients in this section. However, the results are interesting. We have consulted with specialists in the area, and they consider that the information can be useful to understand the behavior of the pattern of use of the molecule and to facilitate clinical decision-making. In this sense, we consider it prudent to maintain these sections.

In the discussion:

- What do you mean by this sentence "The use of mycophenolate mofetil, especially in women, is related to the frequency of SLE (16) diagnosis".

R/SLE is more common in women than in men. In our study, most of the population consisted of women, which may be related to the fact that SLE was the main documented indication for mycophenolate mofetil. We have added a sentence in this regard.

- This was not reflected in the results "The use of angiotensin-converting enzyme inhibitors or angiotensin 2 receptor antagonists is highly recommended, as identified in this report, as is the use of sodium-glucose 2 cotransporter inhibitors", only in the table. Also, no explanation was provided.

R/ This phrase refers to the previous sentence regarding lupus nephritis. Patients with (or at risk of) this condition are recommended to be prescribed ARBs or ACEis. These two groups of drugs were commonly prescribed to the study population, as shown in Table 3. However, to assess this in more detail, we have added new columns in Table 3 for SLE and solid organ transplant.

- I did not see in the results "The use of mycophenolate mofetil for solid organ transplantation, in this case, frequently liver transplantation has also been reported" and I did not understand how you reached it.

R/ This phrase was rewritten to point out that the drug has been described in patients with organ transplantation, using liver transplantation as an example, not necessarily focusing on the obtained results.

- How is this relevant "The use of mycophenolate mofetil for solid organ transplantation, in this case frequently liver transplantation, has also been reported in the United Kingdom (20), as the recommended dose is approximately 1000 mg every 12 hours, usually in combination with a calcineurin or glucocorticoid inhibitor (21). The efficacy of mycophenolate mofetil for this purpose has also been demonstrated compared to treatments based on calcineurin inhibitors to avoid graft rejection and patient survival for up to 5 years of follow-up (22). "?

R/ The last sentence mentioned in this commentary was shortened to make it more concise. The treatment is now referred to as mycophenolate monotherapy. The other phrases in the same paragraph have also been revised accordingly.

- The discussion section needs critical revision to improve the discussion level.

R/ The discussion section has been revised in accordance with the other recommendations from the reviewers.

Comments on the Quality of English Language

The manuscript has many English writing errors and needs revision by an editing service.

R/ We have asked for a second revision by the editing service that originally reviewed the document. We attach the new AJE certificate.

- Disease Modifying Antirheumatic Drugs (DMARDs), the term was defined many times in the manuscript which is against the scientific writing standards.

R/ It was duplicated and now it is only stated the first time in the methods. However, the term is also explained in each table where necessary according to the standards for scientific writing.

The conclusion also needs revision to reflect the study's purpose.

R/ We disagree, as the conclusion states the broad pattern of mycophenolate use, which was the study objective. However, we have performed several edits to this section to clarify the conclusions.

We hope to have responded successfully to all of the reviewer comments. Thank you for your consideration.

The authors

Reviewer 2 Report

Comments and Suggestions for Authors

I read with interest the paper titled "Prescription patterns of mycophenolate mofetil in a group of patients from Colombia"

The article is well written, I have minor comments to add. 

1. Please explain further in the background why is expected such high percentage of women, or why in your sample there are too many. 

2. Is being an adult an inclusion criteria? Please clarify inclusion and exclusion criteria. 

3. The duration of use was not available?

4. As a major suggestion, the analysis of subgroups could be extended of analyzing the main indications independently of the other cases (eg. SLE vs others), because the pattern of use in different diseases will be clearly different from other (as an example, rheumatic diseases will have different comcomitant drugs as DMARDs, that are usually not present in other indications, such as transplants or cardiovascular pathologies. 

Author Response

Response to reviewer 2

Manuscript ID: healthcare-3161455

Title: Prescription patterns of mycophenolate mofetil in a group of patients from Colombia

Thank you for considering our paper. Here, we respond point by point to the comments

I read with interest the paper titled "Prescription patterns of mycophenolate mofetil in a group of patients from Colombia".

The article is well written, I have minor comments to add.

  1. Please explain further in the background why is expected such high percentage of women, or why in your sample there are too many.

R/ The sentence regarding this topic has been expanded in the second paragraph of the discussion. This can be explained by the high frequency of SLE, which is more common in women.

  1. Is being an adult an inclusion criteria? Please clarify inclusion and exclusion criteria.

R/ Yes. We have now specifically indicated that we included only adults; this criterion can be found in the second paragraph of the methods section.

  1. The duration of use was not available?

R/ No, we do not have data regarding the duration of use. We have added this information to the limitations paragraph.

  1. As a major suggestion, the analysis of subgroups could be extended of analyzing the main indications independently of the other cases (eg. SLE vs others), because the pattern of use in different diseases will be clearly different from other (as an example, rheumatic diseases will have different comcomitant drugs as DMARDs, that are usually not present in other indications, such as transplants or cardiovascular pathologies.

R/ We have now included supplementary tables with the same structure as that of Table 2 but divided by indications.

We hope to have responded successfully to all of the reviewer comments. Thank you for your consideration.

The authors

Reviewer 3 Report

Comments and Suggestions for Authors

General comments:

·       While this only cross-sectional study, no inferences in terms associations between various characteristics of patients and treatments were made

·         The study may be limited in terms of generalizability, relevance, and overall value to clinical practice

·         The manuscript is generally well written, it is filled with run-on sentences, which could be addressed.

·         The research question of this paper was clearly stated as the description of the patterns of use of mycophenolate mofetil in patients in Colombia.

·         The study population was clearly specified and defined but I am somewhat surprised that only 979 adults were prescribed mycophenolate out of 9.2 million. I am wondering if other exclusion criteria were not used that were not clearly stated in the article. In general, the timeframe was reasonable.

Specific Suggestions:

·         On line number 41:“It is eliminated mainly in the urine…” I suggest to make it more clear.

·         Please consider the following suggestion,  “Mycophenolate mofetil is primarily eliminated through the urine.”

·         On line numbers 48-51: “Studies carried out in the United States have shown evidence of its effectiveness over other drugs, such as glucocorticoids and azathioprine, initially in the prophylaxis of solid organ rejection, and based on these efficacy results, it has become the leading drug for this purpose…” This is a run on sentence. Split into 2 sentences. Please consider the following suggestion,  “… Based on these efficacy results, it has become the leading drug for this purpose”

·         On line numbers 53-59: “Being considered relatively safe with minor or manageable toxicity and can be associated with some cardiovascular reactions, such as chest pain, hypertension, edema, and tachycardia, in patients requiring high doses, in addition to anxiety, hyperglycemia, altered lipid and electrolyte profiles, abdominal pain, anorexia, diarrhea, urinary tract infections, urticaria, anemia, leukopenia, altered liver enzymes, increased incidence of infections, candidiasis, sepsis, increased creatinine, cough, dyspnea, and fever…” This is a run on sentence.  Consider changing to 3 sentences for clarity Please consider the following suggestion, “Being considered relatively safe with minor or manageable toxicity. However, in patients requiring high doses, it can be associated with various cardiovascular reactions, such as chest pain, hypertension, edema, and tachycardia. Other potential side effects include anxiety, hyperglycemia, altered lipid and electrolyte profiles, abdominal pain, anorexia, diarrhea, urinary tract infections, urticaria, anemia, leukopenia, altered liver enzymes, an increased incidence of infections (including candidiasis and sepsis), increased creatinine levels, cough, dyspnea, and fever.”

·         On line numbers 79-85: “A cross-sectional study was conducted on the prescription patterns of mycophenolate mofetil (Micoflavin®) from a drug dispensing database that recorded information on an approximate population of 9.2 million people affiliated with four different insurers of the Colombian Health System, who make up 17.3% of the country's population, 85% of them belonging to the contributory regime and the other 15% to the subsidized scheme (correspond to people without formal work, who are covered in the health system by the state).” These are run on sentences. Please consider the following suggestion, “A cross-sectional study was conducted to analyze the prescription patterns of mycophenolate mofetil (Micoflavin®) using a drug dispensing database. This database recorded information on an approximate population of 9.2 million people affiliated with four different insurers within the Colombian Health System. This population represents 17.3% of the country's total population, with 85% enrolled in the contributory regime and the remaining 15% in the subsidized scheme, which covers individuals without formal employment who are supported by the state.?

·         On line numbers 95-96: “…, as follows: Caribbean Region, Central Region, BogotáCundinamarca, Pacific Region and Amazon Region, and Orinoquía-Oriental Region.” This is an run on sentence. Consider making this into a separate sentence

·         Please consider the following suggestion, “The categories are as follows: Caribbean Region, Central Region, Bogotá-Cundinamarca, Pacific Region, Amazon Region, and Orinoquía-Oriental Region.”

·         For line 105: I am not sure that “pharmacological” is an appropriate title. This subsection seems to be limited to the dose.

·         On line 111: subtitle should be concomitant medications and not just “medications”

·         On line numbers 117 and 170: DMARD Use of repeated non- abbreviations, should have been done on 117 and not 170. Disease Modifying Antirheumatic Drugs (DMARDs)

·         On line numbers 156-160: Table 2 shows the variables related to the patterns of use, such as the average dose, dosage ranges, and administration intervals, according to the two presentations of the drug used (250 and 500 mg) for patients overall and according to whether the primary diagnosis was SLE or a solid organ transplant. The 92.2% (n=903) used the drug twice daily, and only 7.8% once daily. This description sounds as if it should be a part of the table and not in the article?

·         On line number 182: “an average of 7.2 prescribed drugs were found…” Numbers, should this be rounded to a whole number?

·         On line numbers 190-193: “DMARD: Disease Modifying Antirheumatic Drug; ARNI: Angiotensin Receptor – Neprilysin Inhibitor; DPP4: Dipeptidyl peptidase 4; GLP-1: Glucagon-like peptide; SGLT2i: Sodium glucose co-transporter type 2 inhibitor. a: Other biological DMARD: abatacept, adalimumab, anakinra, certolizumab, etanercept, golimumab, hydroxyurea, infliximab, tocilizumab, ustekinumab.”

·         There is an issue with placement. Not sure if this is in the proper spot. Unsure where this is suppose to be.

·         On line number 214: “SLE (16) diagnosis…” Is this a reference? Usually the sentence

·         On line numbers 280-288: “According to the mentioned findings, mycophenolate mofetil was mainly indicated as a combination therapy in patients with SLE, patients with other rheumatological diseases and patients with solid organ transplants (liver and kidney) at doses recommended by international guidelines for the main indications, with a high frequency of concomitant prescription of antihypertensives, statins, proton pump inhibitors, and other immunomodulators and DMARDs. These results are a first approximation of the patterns of use of this therapy in Colombia, opening the need for new studies to identify relevant information, such as persistence of use, adherence to therapy, and complications, and potentially broaden the panorama of the use of mycophenolate mofetil.” Run on sentences, didn’t need to spell out abbreviations on SLE or DMARDS. Please consider the following suggestion, “According to the findings, mycophenolate mofetil was primarily prescribed as part of combination therapy for patients with systemic lupus erythematosus (SLE), other rheumatological diseases, and solid organ transplants (liver and kidney). The doses used were consistent with international guidelines for these main indications. There was a high frequency of concomitant prescription of antihypertensives, statins, proton pump inhibitors, and other immunomodulators and disease-modifying antirheumatic drugs (DMARDs). These results provide an initial overview of mycophenolate mofetil usage patterns in Colombia and highlight the need for further studies to gather additional information on aspects such as therapy persistence, adherence, complications, and to potentially expand the understanding of mycophenolate mofetil use.”

Comments on the Quality of English Language

See above

Author Response

Response to reviewer 3

Manuscript ID: healthcare-3161455

Title: Prescription patterns of mycophenolate mofetil in a group of patients from Colombia

Thank you for considering our paper. Here, we respond point by point to the comments

General comments:

  •      While this only cross-sectional study, no inferences in terms associations between various characteristics of patients and treatments were made

R/ This was not the scope of the study; it was intended to present only descriptive analyses.

  • The study may be limited in terms of generalizability, relevance, and overall value to clinical practice.

R/ Yes, we state this caveat in the limitations section. However, we also present the strengths of the study in the same paragraph.

  • The manuscript is generally well written, it is filled with run-on sentences, which could be addressed.

R/ We have revised these run-on sentences in consideration of the reviewers’ comments. We have also asked for a second review from the editorial service provider.

  • The research question of this paper was clearly stated as the description of the patterns of use of mycophenolate mofetil in patients in Colombia.

R/ NA

  • The study population was clearly specified and defined but I am somewhat surprised that only 979 adults were prescribed mycophenolate out of 9.2 million. I am wondering if other exclusion criteria were not used that were not clearly stated in the article. In general, the timeframe was reasonable.

R/ All the criteria are shown in the manuscript. This number of patients may be explained by the fact that this study was conducted with data from only one brand of the drug, which is expected to represent approximately 25% of the mycophenolate mofetil formulations available in our country.

Specific Suggestions:

  • On line number 41:“It is eliminated mainly in the urine…” I suggest to make it more clear. Please consider the following suggestion: “Mycophenolate mofetil is primarily eliminated through the urine.”

R/ This has been revised as suggested. Thank you.

  • On line numbers 48-51: “Studies carried out in the United States have shown evidence of its effectiveness over other drugs, such as glucocorticoids and azathioprine, initially in the prophylaxis of solid organ rejection, and based on these efficacy results, it has become the leading drug for this purpose…” This is a run on sentence. Split into 2 sentences. Please consider the following suggestion,  “… Based on these efficacy results, it has become the leading drug for this purpose”

R/ We have shortened this sentence according to a comment from another reviewer.

  • On line numbers 53-59: “Being considered relatively safe with minor or manageable toxicity and can be associated with some cardiovascular reactions, such as chest pain, hypertension, edema, and tachycardia, in patients requiring high doses, in addition to anxiety, hyperglycemia, altered lipid and electrolyte profiles, abdominal pain, anorexia, diarrhea, urinary tract infections, urticaria, anemia, leukopenia, altered liver enzymes, increased incidence of infections, candidiasis, sepsis, increased creatinine, cough, dyspnea, and fever…” This is a run on sentence.  Consider changing to 3 sentences for clarity Please consider the following suggestion, “Being considered relatively safe with minor or manageable toxicity. However, in patients requiring high doses, it can be associated with various cardiovascular reactions, such as chest pain, hypertension, edema, and tachycardia. Other potential side effects include anxiety, hyperglycemia, altered lipid and electrolyte profiles, abdominal pain, anorexia, diarrhea, urinary tract infections, urticaria, anemia, leukopenia, altered liver enzymes, an increased incidence of infections (including candidiasis and sepsis), increased creatinine levels, cough, dyspnea, and fever.”

R/ We have revised this paragraph accordingly. We have also shortened some sentences.

  • On line numbers 79-85: “A cross-sectional study was conducted on the prescription patterns of mycophenolate mofetil (Micoflavin®) from a drug dispensing database that recorded information on an approximate population of 9.2 million people affiliated with four different insurers of the Colombian Health System, who make up 17.3% of the country's population, 85% of them belonging to the contributory regime and the other 15% to the subsidized scheme (correspond to people without formal work, who are covered in the health system by the state).” These are run on sentences. Please consider the following suggestion, “A cross-sectional study was conducted to analyze the prescription patterns of mycophenolate mofetil (Micoflavin®) using a drug dispensing database. This database recorded information on an approximate population of 9.2 million people affiliated with four different insurers within the Colombian Health System. This population represents 17.3% of the country's total population, with 85% enrolled in the contributory regime and the remaining 15% in the subsidized scheme, which covers individuals without formal employment who are supported by the state.?

R/ This text has been revised as suggested. Thank you.

  • On line numbers 95-96: “…, as follows: Caribbean Region, Central Region, BogotáCundinamarca, Pacific Region and Amazon Region, and Orinoquía-Oriental Region.” This is an run on sentence. Consider making this into a separate sentence. Please consider the following suggestion, “The categories are as follows: Caribbean Region, Central Region, Bogotá-Cundinamarca, Pacific Region, Amazon Region, and Orinoquía-Oriental Region.”

R/ These lines have also been revised.

  • For line 105: I am not sure that “pharmacological” is an appropriate title. This subsection seems to be limited to the dose.

R/ We have changed this term in the title to “medication”.

  • On line 111: subtitle should be concomitant medications and not just “medications”

R/ The subtitle has been changed.

  • On line numbers 117 and 170: DMARD Use of repeated non- abbreviations, should have been done on 117 and not 170. Disease Modifying Antirheumatic Drugs (DMARDs)

R/ Corrected.

  • On line numbers 156-160: Table 2 shows the variables related to the patterns of use, such as the average dose, dosage ranges, and administration intervals, according to the two presentations of the drug used (250 and 500 mg) for patients overall and according to whether the primary diagnosis was SLE or a solid organ transplant. The 92.2% (n=903) used the drug twice daily, and only 7.8% once daily. This description sounds as if it should be a part of the table and not in the article?

R/ We believe that it is not necessary to include the information from the last sentence in the table.

  • On line number 182: “an average of 7.2 prescribed drugs were found…” Numbers, should this be rounded to a whole number?

R/ This number has been revised according to the recommendation.

  • On line numbers 190-193: “DMARD: Disease Modifying Antirheumatic Drug; ARNI: Angiotensin Receptor – Neprilysin Inhibitor; DPP4: Dipeptidyl peptidase 4; GLP-1: Glucagon-like peptide; SGLT2i: Sodium glucose co-transporter type 2 inhibitor. a: Other biological DMARD: abatacept, adalimumab, anakinra, certolizumab, etanercept, golimumab, hydroxyurea, infliximab, tocilizumab, ustekinumab.”. There is an issue with placement. Not sure if this is in the proper spot. Unsure where this is suppose to be.

R/ These lines are supposed to compose the table footnotes. We now use a smaller font size to differentiate this text from the rest of the manuscript.

  • On line number 214: “SLE (16) diagnosis…” Is this a reference? Usually the sentence

R/ We have moved this citation to the end of the sentence.

  • On line numbers 280-288: “According to the mentioned findings, mycophenolate mofetil was mainly indicated as a combination therapy in patients with SLE, patients with other rheumatological diseases and patients with solid organ transplants (liver and kidney) at doses recommended by international guidelines for the main indications, with a high frequency of concomitant prescription of antihypertensives, statins, proton pump inhibitors, and other immunomodulators and DMARDs. These results are a first approximation of the patterns of use of this therapy in Colombia, opening the need for new studies to identify relevant information, such as persistence of use, adherence to therapy, and complications, and potentially broaden the panorama of the use of mycophenolate mofetil.” Run on sentences, didn’t need to spell out abbreviations on SLE or DMARDS. Please consider the following suggestion, “According to the findings, mycophenolate mofetil was primarily prescribed as part of combination therapy for patients with systemic lupus erythematosus (SLE), other rheumatological diseases, and solid organ transplants (liver and kidney). The doses used were consistent with international guidelines for these main indications. There was a high frequency of concomitant prescription of antihypertensives, statins, proton pump inhibitors, and other immunomodulators and disease-modifying antirheumatic drugs (DMARDs). These results provide an initial overview of mycophenolate mofetil usage patterns in Colombia and highlight the need for further studies to gather additional information on aspects such as therapy persistence, adherence, complications, and to potentially expand the understanding of mycophenolate mofetil use.”

R/ This text has been revised according to the recommendation. Thank you.

We hope to have responded successfully to all of the reviewer comments. Thank you for your consideration.

The authors

Round 2

Reviewer 1 Report

Comments and Suggestions for Authors

The authors need to review the comments and try to address them comprehensively.

Comments on the Quality of English Language

Still need revision

Author Response

We send the manuscript to American Journal Experts to edit english languaje.

We attached the certificate

The authors
